# Looking beyond the Lamppost: Population-Level Primary Prevention of Breast Cancer

**DOI:** 10.3390/ijerph17238720

**Published:** 2020-11-24

**Authors:** Gabriella M. McLoughlin, Eric M. Wiedenman, Sarah Gehlert, Ross C. Brownson

**Affiliations:** 1Implementation Science Center for Cancer Control and Prevention Research Center, Brown School, Washington University in St. Louis, One Brookings Drive, St. Louis, MO 63130, USA; ericw@wustl.edu (E.M.W.); rbrownson@wustl.edu (R.C.B.); 2Department of Surgery, Division of Public Health Sciences, Washington University School of Medicine, Washington University in St. Louis, St. Louis, MO 63110, USA; 3Suzanne Dworak-Peck School of Social Work, the University of Southern California, Los Angeles, CA 90089, USA; gehlert@usc.edu; 4Desmond Lee Professor of Racial and Ethnic Diversity, Brown School, Washington University in St. Louis, One Brookings Drive, St. Louis, MO 63130, USA

**Keywords:** breast cancer, dissemination and implementation, policy, translational research

## Abstract

Although innovative and impactful interventions are necessary for the primary prevention of breast cancer, the factors influencing program adoption, implementation, and sustainment are key, yet remain poorly understood. Insufficient attention has been paid to the primary prevention of breast cancer in state and national cancer plans, limiting the impact of evidence-based interventions on population health. This commentary highlights the state of primary prevention of breast cancer and gaps in the current literature. As a way to enhance the reach and adoption of cancer prevention policies and programs, the utility of dissemination and implementation (D&I) science is highlighted. Examples of how D&I could be applied to study policies and programs for chronic disease prevention are described, in addition to needs for future research. Through application of D&I science and a strong focus on health equity, a clearer understanding of contextual factors influencing the success of prevention programs will be achieved, ultimately impacting population health.

## 1. Introduction

Primary prevention of breast cancer is a significant public health concern [1]. Breast cancer is the second most common form of cancer for adults in the United States (USA), with 250,000 women diagnosed every year [2]. Worldwide, breast cancer is the most common among women; over 2 million women were diagnosed in 2018 [3]. Within the USA and globally, the disproportionate risk of breast cancer mortality for individuals of low-income and racial/ethnic minorities warrants more innovative prevention approaches grounded in health equity and social justice [4,5]. Although overall prevalence has not significantly increased in the last few years, incidence of breast cancer in the US has increased for Black and Asian/Pacific Islander women slightly higher than for White women, with a 43% higher rate of mortality among Black women than White women [2]. Accordingly, a growing body of research has manifested to address the social, behavioral, and environmental factors that contribute to growing inequities in deaths from breast cancer, emphasizing not only focus on increasing survivorship but also prevention of incidence [6,7,8].

Primary prevention of breast cancer has not often been a focus of state cancer control plans, although there is considerable overlap of risk factors with other cancer sites [9], highlighting the need for improved prevention strategies that help mitigate disparities in risk for cancer and overall health of those most disadvantaged. This may reflect the general trend toward treatment over prevention in medicine, specifically within oncology [10]. Numerous strategies have been recommended to reduce the risk for breast cancer (i.e., reducing alcohol consumption, increasing physical activity, healthy diet) [11], but such behaviors are encapsulated by multiple environmental/contextual issues, such as structural inequalities, physical environment, food security, policies concerning air and water safety, and chemical use [6,12,13]. Such factors influence an individual’s ability to adopt health-promoting habits, thus hindering their protective impact on prevalence risk [14]. Moreover, associations among income level and genomic expression of cancer cells indicate that those identified as low income are more likely to experience complications with cancer treatment [15,16,17]. One example of a federally-funded program to address these issues is the National Breast and Cervical Cancer Early Detection Program (NBCCEDP) [18], which has provided screening services to low-income, uninsured, and underserved women. Programs such as these mark a vital service in offsetting health disparities for breast cancer detection and prevention, yet little is known regarding how such programs are implemented and their impact on reducing disparities in prevalence. Empirical research to identify effective programs and policies that mitigate risks for disadvantaged populations, in addition to ways such programs can be effectively implemented, is therefore necessary [19,20].

In this commentary, we present a brief summary of the literature regarding primary prevention of breast cancer, identifying gaps in the literature and avenues for innovation. Further, we introduce potential opportunities for dissemination and implementation (D&I) science as a means to enhance our understanding of the linkages (and gaps) between policy, research, and practice, providing a pragmatic lens through which to frame current and future intervention work.

## 2. Primary Prevention of Breast Cancer: A Brief Review of Reviews

Much of the breast cancer prevention research has focused on identifying factors associated with higher or lower rates of breast cancer diagnosis, followed by interventions designed to impact the factors most modifiable. Table 1 summarizes key risk factors for breast cancer. Environmental and behavioral factors are associated with 90 to 95 percent of breast cancer cases [6,13]. Behavioral risk factors have been given considerable attention in the literature [1,11,13,21,22,23,24,25], with healthy diet (avoiding high calorie foods, processed meat, and alcohol consumption) and weight management as the most often discussed.

With regard to environmental risk factors, psychological factors, such as stress, anxiety, and depression have been linked to breast cancer incidence in systematic review research, indicating that individuals who report greater prevalence of these symptoms are also at a greater risk for diagnosis [26,27]. However, these associations were found in rather heterogeneous populations and should be interpreted with caution, particularly the relative degree of influence acute versus chronic stress/anxiety/depression has on risk for breast cancer [26,27]. Another noteworthy environmental risk factor is the role of shift work, particularly working late at night, as a risk factor for breast cancer [28]. Dysregulation of melatonin and disrupted sleep schedule, predominantly observed in low-income women working shift patterns, have been shown to increase risk for cancer [6]. In addition, a review by Brody et al. [29] highlighted that increased risk for breast cancer was associated with exposures to high levels of total suspended particulates, gasoline and vehicular exhaust, and polychlorinated biphenyls (PCBs). These particulates are increasing as a consequence of climate change. Other chemical exposures identified to increase breast cancer incidence include dichlorodiphenyltrichloroethane (DDT; pesticide), polycyclic aromatic hydrocarbons (PAH; produced through combustion of hydrocarbons), and endocrine-disrupting chemicals (EDC; lead, chlorine, arsenic) [29,30,31]. Accordingly, it is recommended that reducing environmental exposures be a public health priority as a means to decrease risk for breast cancer incidence [12,29].

Further examination of socio-environmental exposures and their impact on breast cancer incidence has identified specific windows of susceptibility (WOS): biological landmarks when a woman’s breast tissue changes including the prenatal, pubertal, pregnancy, and menopausal periods [31,32]. Indeed, WOS are often associated with significant developmental or life-event circumstances that may make targeted interventions more effective. Middle adulthood, for example, may provide more opportunities for health behavior change and increased effectiveness of behavioral interventions aimed at reducing breast cancer risk [33]. This time period may be of particular significance, as the risks associated with various behaviors (alcohol consumption, tobacco use, exposure to light at night, sleep loss, and physical inactivity) begin to impact physical health. Moreover, about 80 percent of breast cancer cases in the United States occur in women 50 years of age or older [34], highlighting the importance for interventions during midlife and earlier windows of susceptibility. This age range also represents a potentially important period for secondary prevention through screening measures. High-penetrance genes, such as BRCA1 and 2, are one of the strongest risk factors for breast cancer incidence [14], and screening of women over 50 and those at high risk results in a 30 percent reduction in cancer mortality [35]. Further, individuals with a BRCA1/2 genetic risk of breast cancer can reduce their breast cancer incidence by 50 percent through a salpingo-oophorectomy [35]. Therefore, WOS generally and middle adulthood specifically involve complex interactions between multiple levels of risk factors [4], emphasizing the need for multi-level, population-based cancer prevention interventions.

Other environmental and social determinants impact breast cancer incidence and mortality, with socioeconomic status and educational opportunity found to be driving factors of both [13]. Education, income, and ancestry are strongly associated with risk factors such as age at first birth, parity, breast feeding, and tobacco and alcohol use [13]. Further evidence suggests income is a strong determinant of cancer survivorship and low socioeconomic status is related to difficulty finding/affording healthy foods and is connected to poor cancer treatment outcomes, including increased risk of death from treatable cancers [35]. In addition to aforementioned psychological factors of depression/anxiety and stress [26,27], racial/ethnic minorities also face discrimination, lack of social support, and social isolation, which are found to be related to breast cancer stage at diagnosis as well as chances of survival [36] (Table 1). Accordingly, future research efforts should focus on improving the implementation of evidence-based interventions to ensure that all individuals within a community benefit from cancer prevention and early detection intervention programming [35,37,38]. However, a significant need still exists to understand the social, political, and physical environmental factors that can be used to increase implementation of these interventions, specifically in underserved and racial/ethnic minority populations [35].

Brownson and colleagues [39] summarized key components of population-level prevention for chronic disease and for their utility within implementation science. The authors highlight seven points: (i) environmental and policy interventions are the key to initiating and maintaining systematic change; (ii) consider multiple levels of influence; (iii) make better use of existing tools for implementation; (iv) understand local context and politics; (v) build new and nontraditional relationships; (vi) address health disparities; and (vii) conduct more and better policy research. The Institute of Medicine (IOM) (now the National Academy of Medicine) also addressed this first point through their recommendations of public health and systems approaches to interventions focused on the physical activity, food and beverage, and school work environments [40]. Thus, future investigators should understand the context, value system, socioeconomic status, unique demographics, and other factors within the communities they are engaged in prior to intervention development [33].

## 3. Adopting a Health Equity Lens to Cancer Prevention

Grounded in addressing social determinants of health, a key emphasis is on stakeholder (i.e., end-user) engagement through finding ways to reach at-risk individuals within community settings, enhancing quality of care. For example, Hardy and Bugella implemented a community-based intervention that targeted college students and peri- and post-menopausal women in Alabama through places of worship, highlighting the meaningful intersection of race, gender, and faith as key factors impacting breast cancer [42], particularly in Black communities. In addition, Salant and Gehlert conducted focus group interviews with individuals in Black communities, which provided valuable insights on their perceptions and internalizations of breast cancer risks, and the work still to be done to earn community trust in researchers and medical professionals [5,49]. Such inquiry revealed the gaps in knowledge dissemination and lack of awareness regarding available resources for cancer prevention and treatment [49].

In addition to gathering stakeholder input, researchers should expand their inquiry to explore the multiple levels of influence impacting breast cancer prevention [4], with environmental and policy interventions identified as having the most impact and sustainability for systemic change [39]. Breast cancer incidence is linked to multiple social determinants of health such as race/ethnicity, income, and living conditions [6,9,13,33,35], and a successful intervention should work to address these determinants. For example, rates of breast cancer for black women under 50 are significantly greater than for White women [6], highlighting the need to target primary prevention efforts at this subgroup at an earlier life stage. Interventions should be sensitive to the unique cultural, societal, and historical context within the community they are serving. As previously highlighted, identifying and rooting an intervention within the lived experiences of disadvantaged populations (e.g., low-income, ethnic minorities, non-college educated) can facilitate patient engagement and make a significant impact on patient/participant health [42,49].

As a response to the growing inequities contributing to high cancer incidence in the US, the National Institutes of Health (NIH) sponsored several centers for population health and health disparities across the country with a key goal of enhancing stakeholder-driven research and combining basic and social sciences to address key issues in population health. Such an approach highlights the need to focus on how distal factors, such as social conditions and policies, interact with those more proximal, such as biological/genetic pathways and individual risk behaviors, contributing to cancer risk [4]. Through their work conducting interviews with content experts and stakeholders, Bridges and colleagues propose a conceptual framework with an equity focus that can be used to foster both global and local action towards breast cancer prevention [51]. Their framework identifies four key areas of action: (i) building capacity within the health care field; (ii) promoting patient advocacy; (iii) developing evidence specific to local etiology; and (iv) removing barriers to care such as cost, disparities in access, and facilitating early detection and reimbursement. Similarly, the Getting to Equity (GTE) in Obesity Prevention Theoretical Framework [52,53] provides researchers with a guideline for increasing the equity effect of Policy, System, and Environmental (PSE) change interventions that aim to decrease obesity. Although designed to apply an equity lens to obesity-related PSE research, it is recommended for all areas of public health research that are relevant to the obesity epidemic in the U.S. such as cancer prevention. The framework identifies four key opportunities for increasing the equity impact of obesity research: (1) increase healthy options, (2) reduce deterrents, (3) improve social and economic resources, and (4) build on community capacity. Such components align somewhat with that of Bridges et al. [51] and take an ecological approach to health promotion rather than focusing on individual behaviors.

Implementation of these health equity frameworks for cancer prevention would require dedicated resources (i.e., financial, political, and scientific communication support) and stakeholder engagement from diverse groups within multiple ecological layers of influence, such as clinicians, community-based program officers, researchers, and patients [51]. Further, scant knowledge regarding how such complex systems-level interventions are developed and implemented in practice, and the factors influencing adoption and implementation, provides rationale for integration of D&I science in primary prevention efforts. Below, we discuss ways in which the field of primary prevention may benefit from translational research efforts through D&I principles, highlighting areas for potential research and practice.

## 4. The Role of Dissemination and Implementation Science in Primary Prevention

To understand how policies and practices for the primary prevention of breast cancer can have the most potent effect, it is crucial to assess the factors leading to successful adoption and implementation of effective interventions. Dissemination and implementation (D&I) science is the study of the adoption of evidence-based practices when introduced to a new setting or population to minimize the research-to-practice gap [54,55,56,57]. A strength of D&I is the emphasis on a stakeholder-driven approach in all facets of the research process through soliciting the needs and interests of those tasked with implementation and can be integrated as part of community-based participatory research (CBPR) [58]. Through such practitioner-focused research, tailored implementation strategies may be developed based on input from key stakeholders both within and outside clinical settings to enhance the likelihood that a particular policy or practice will be utilized [59,60,61]. Below, we provide a case study example of how D&I research may be applied to primary breast cancer prevention through the California Breast Cancer Research Program (CBCRP), based on key principles of the program [9].

## 5. Case Study Example: California Breast Cancer Research Program

As a cross-disciplinary group of academics, policy makers, and providers, the CBCRP, in collaboration with Breast Cancer Prevention Partners, is working to develop a comprehensive breast cancer primary prevention plan to serve as a roadmap for prevention research and practice [9]. The plan is currently under development through an iterative community-based participatory process by gaining insights of stakeholders from several “listening” sessions held throughout California. In these sessions, key stakeholders (i.e., public health professionals, clinicians, and community workers) shared their needs and experiences related to breast cancer primary prevention. Main themes from these sessions addressed the need for systemic, policy, and environmental change as a key priority, superseding the emphasis on individual behavior change [9] for primary prevention. As such, the plan addresses actions at the state, county, and local levels, which may be implemented to mitigate risk for breast cancer and emphasize a multifaceted and comprehensive approach, grounded in social justice [8,9].

As previously identified, there is a lack of clear evidence pointing to effective population-level programs and/or policies for primary prevention of breast cancer when examining a wide range of risk factors [1]. Given the distinct inequities faced by low-income and racial/ethnic minority populations, it is therefore necessary to develop and test system-level policies and interventions through a D&I lens, as opposed to more traditional randomized controlled trials which offer limited external validity [62]. For example, health behavior incentives, such as the Double Up Food Bucks Program (DUFB), were developed as systems-level policy interventions with the goal of enhancing chronic disease prevention behaviors, such as cancer, through incentivizing the purchase and consumption of fruits and vegetables [63]. Through such an incentive, individuals who are eligible for Supplemental Nutrition Assistance Programs (SNAP) can purchase fruits and vegetables from any participating grocery store or farmers market and receive twice as much produce for their money. Overall, the incentive program has been well-received and shown to be effective at facilitating the purchase of fruits and vegetables within low-income populations [64,65]; barriers to participation are lack of transportation/difficulty accessing location and low awareness in eligible populations.

Based on extant research, a potential policy intervention at the state and county level could be to examine how county public health departments/and community health centers can enhance awareness and accessibility of DUFP for eligible participants. Counties may be tasked with (a) increasing awareness through a multicomponent communication strategy using social media, email reminders, and/or solicitation at community centers and clinics; and (b) increasing access by facilitating the use of public transport or supporting placement of DUFP partners in food deserts and/or fast food “swamps.” Following initial development and pilot implementation, county public health officials could then report to state public health departments on strategies that were effective and feasible to implement, thus generating a list of potential implementation strategies [59,60,61] based on key stakeholder input. Such practitioner-focused research has been recommended by D&I scholars as a way to enhance external validity and translatability of findings so they can be rapidly put in to practice [62,66].

From these initial best practices and pilot research, leadership and policy support from the state-level public health office could partner with community stakeholders, such as the community-based organizations affiliated with CBCRP. Such collaborative relationship would facilitate implementation of systems-level health incentive programs, such as DUFP, and empower counties to refine implementation strategies based on the nuanced needs and challenges of their communities. Data sharing practices could also be applied whereby county-level officials report the rates of participation in incentive programs and the particular strategies used for policy implementation, so that information can be gleaned about implementation effectiveness [67], thereby facilitating rapid modification and adaptation of program implementation. Through such an iterative process, this continuous plan-do-study-act cycle [68] will facilitate ongoing refinement of policies and programs that shape health behaviors of vulnerable and marginalized populations at greatest risk for breast cancer. Similar approaches can be taken with other policy and/or incentive programs that strive to reduce barriers to accessing healthcare and fostering health behaviors.

## 6. Implications for Primary Prevention and Practice

Our brief commentary summarized the current evidence surrounding primary prevention of breast cancer and identified opportunities for D&I science to enhance the uptake of evidence to practice and policy. Efforts are already underway to integrate community-based participatory research and D&I into evidence-informed interventions across California as a means to integrate stakeholder involvement in rapid-cycle research [69]. Such work offers myriad advantages to understanding how local- and state-level policies and evidence-based interventions can be adopted and integrated into routine practice. Researchers and health professionals can examine implementation effectiveness to see if counties adopt more systems-level thinking and policy, and the impact on cancer incidence in their populations over time. This evaluation can be conducted using practitioner-focused research tools (e.g., data sharing of medical records/screening, check-in surveys, and focus group interviews) to rapidly adapt implementation plans depending on county-specific needs. We recognize that breast cancer prevention in California and the USA is not representative of other countries, and this geographic variation should be considered when applying D&I concepts to other countries. Therefore, future research is needed to investigate nationwide screening/prevention efforts in low, middle, and high-income countries and address their impact on incidence and treatment. Finally, D&I experts working with the CBCRP [9,69] can provide training to county-level public health officials regarding best practices for data sharing and fostering collaborations/information sharing to facilitate understanding of policy implementation, cancer rates and prevention strategies. These outlined opportunities will require significant financial and scientific support to reduce risk for breast cancer; however, such an investment in translational research and systems change yields positive implications for population health and health equity.

## 7. Conclusions

This commentary summarizes the current status and further needs pertaining to primary prevention of breast cancer, and the potential utility for dissemination and implementation (D&I) science in such prevention efforts. Given the social determinants of health which disproportionately affect those identified as racial/ethnic minority and with low-income, policy and systems-level innovations are necessary. Although many interventions and policies can be enacted to address social determinants of health, it is crucial to evaluate such initiatives effectively. We provide pragmatic ways that D&I can be integrated to study how prevention policies are adopted, implemented, and sustained, with a key emphasis on stakeholder input. Through such an implementation-focused lens, we hope that evidence-based policies and interventions will have a greater impact on population health and prevention of cancer and chronic disease.

## Figures and Tables

**Table 1 ijerph-17-08720-t001:** Breast cancer risk factors.

Factors	Impact on Breast Cancer (BC) Risk
**Biologic**	
Age	Increase in risk with increase in age [34]
Age at first birth	Older age increases overall risk and breast cancer density risk [34]
Age at menarche	Earlier age increases overall risk [34]
Age at menopause	Later age increases overall risk [34]
Breast density	Increases overall risk [34]
Family history	Increases overall risk [34]
Race/Ethnicity	Strongly associated with behavioral factors in addition to age at first birth, age at menarche, education, income, obesity, tobacco use, and breast density [7,34,36,41,42]
Alcohol consumption	Increases overall risk; identified as leading modifiable risk factor [1,7,23,41,43]
Mediterranean/Low glycemic index diet	Associated with decreased overall risk [24,44,45]
Weight management	Increases overall risk as age increases, especially after menopause [7,45,46]
**Individual/Sociocultural/Environmental**	
Breastfeeding	Decreases overall incidence [23,47]
Parity	Decreases breast density and overall risk [42,47,48,49]
Obesity	Moderately associated with postmenopausal BC risk Inverse or no association for premenopausal BC risk [7,45,46]
Physical activity	Decreases overall risk, as well as risk for obesity [25,47,50]
Tobacco use	Increases overall risk [7]
Exposure to light at night/sleep loss	Increases overall risk through disruption of sleep cycle, hormone release [7]
Total suspended particles (TSP)	Increases overall risk [6,12,29,30,31]
Chemical exposures (gasoline, exhaust, PCB, DDT, PAH, EDC)	Increases overall risk [6,12,29,30,31]
Windows of susceptibility (WOS)	Commonly identified as prenatal, pubertal, pregnancy, and menopausal time frames. Epidemiologic data support that medications, medical conditions, and environmental exposures during these times may increase BC risk [30,31,46]
Education	Higher levels of education are associated with increased breast feeding (protective factor), income, physical activity, and age at first birth, as well as decreased tobacco use [6,13,36]
Socioeconomic status (SES)	Low SES is associated with increased tobacco and alcohol use, obesity, decreased physical activity, and overall poor outcomes for cancer treatment [6,13,36]
Psychological factors	High stress, anxiety and depression are associated with severity of cancer diagnosis as well as potential survivorship rates [26,27]

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
