# Peer review of "Looking beyond the Lamppost: Population-Level Primary Prevention of Breast Cancer"

_ijerph, 2020, doi:10.3390/ijerph17238720_

Round 1

Reviewer 1 Report

The paper deals with the population-level primary prevention of breast cancer. The paper is meant to be a commentary with the aim to highlight the state of the art of the primary prevention of breast cancer and to identify gaps in the literature. I have a few points to remark. 

  • It seems that the topic is developed considering the U.S. as the main geographical referring point. Breast cancer prevention programs are different country by country, and it would be important to situate better the considerations that the authors have proposed. 
  • I would suggest including psychological determinants among the factors associated with higher or lower rates of breast cancer. 
  • One crucial point regarding breast cancer prevention is related to the BRCA1 and BRCA2 gene mutations and the cascade screening among relatives. I would suggest the authors also include this point in their considerations.
  • Implementation of the BC prevention is also determined by the national guidelines for breast screening and examination, which are different country by country for age thresholds and timeline. I would suggest the authors also include this point in their considerations.
  • finally, I may have missed the point, but I am not completely aware of what are the needs for future research. what are the contextual factors which must be studied and better understood? 

Reviewer 2 Report

The commentary presented by McLoughlin et al. is a good match for the special issue of IJERPH in breast cancer prevention. Unlike most cancer prevention strategies that are individual-oriented, the authors focus on the population-level interventions, especially through dissemination and implementation.

A few minor issues need to be addressed.

  • Line 19. Typo. “…, yet he usefulness…”.
  • Table 1 needs to be referenced in the text.

Reviewer 3 Report

The objective of the commentary is to highlight the applicability of implementation science, gaps in current literature and cancer plans in developing strategies for population level prevention of breast cancer. The authors have also emphasized the need to look beyond typically emphasized therapeutic approaches and focus on prevention programs/strategies for successful implementation of D&I science to prevent/reduce incidence rates of global health problems such as cancer. It is interesting commentary however the article can not be considered for publication in the current form for following reasons.

Major concerns:

1) The article is very poorly written and the authors intended message is lost on several ocassions throughout the manuscript. Could authors please carefully review entire manuscript and correct the syntax errors and rephrase the sentences across various sections of the manuscript as needed.

Minor concerns:
1) The authors have mentioned "Environmental and behavioral factors are associated with 90 to 95 percent of breast cancer cases" on Ln # 67 of page # 2. Could authors please cite a reference for this statement or please rephrase the sentence with relevant facts.
2) The subtitle "Implementation science and primary prevention" is included on every page of the manuscript. Could authors please review and remove the aforementioned redundant subtitle.
3) Please replace "he" with "the" on Ln # 19 of page # 1
4) Please rephrase the sentence "Dissemination and implementation (D&I) have been identified as .......impact population-level prevention" as the current sentence structure do not convey the authors intended message correctly on Page # 1.
5) Please replace "studying" with "study" on Ln # 21 of page # 1.
6) Please replace "increased for Black and Asian/Pacific islander women to slightly higher" with "increased in Black and Asian/Pacific islander women slightly higher" on Ln # 33-34 of page # 1.
7) Please replace "healthful foods" with "healthy foods or healthy food choices" on Ln # 101 of Page # 3.
8) Please replace "to accessing" with "to access" on Ln # 236 of page # 6.

Round 2

Reviewer 1 Report

Thank you for your revisions. I feel the manuscript can be accepted for publication in the present form

Reviewer 3 Report

The authors have addressed the concerns and the manuscript could be considered for publication.